# LED-Based Desktop Analyzer for Fat Content Determination in Milk

**DOI:** 10.3390/s23156861

**Published:** 2023-08-01

**Authors:** Anastasiia Surkova, Yana Shmakova, Marina Salukova, Natalya Samokhina, Julia Kostyuchenko, Alina Parshina, Ildar Ibatullin, Viacheslav Artyushenko, Andrey Bogomolov

**Affiliations:** 1Department of Analytical and Physical Chemistry, Samara State Technical University, Molodogvardeyskaya Street 244, 443100 Samara, Russiaa.bogomolov@mail.ru (A.B.); 2Department of Radio Engineering Devices, Samara State Technical University, Molodogvardeyskaya Street 244, 443100 Samara, Russia; 3Art Photonics GmbH, Rudower Chaussee 46, 12489 Berlin, Germany; sa@artphotonics.de

**Keywords:** milk analysis, fat content, optical multisensor systems, RGB sensor, optical spectroscopy, light emitting diode, Arduino Nano, chemometrics

## Abstract

In dairy, there is a growing request for laboratory analysis of the main nutrients in milk. High throughput of analysis, low cost, and portability are becoming critical factors to provide the necessary level of control in milk collection, processing, and sale. A portable desktop analyzer, including three light-emitting diodes (LEDs) in the visible light region, has been constructed and tested for the determination of fat content in homogenized and raw cow’s milk. The method is based on the concentration dependencies of light scattering by milk fat globules at three different wavelengths. Univariate and multivariate models were built and compared. The red channel has shown the best performance in prediction. However, the joint use of all three LED signals led to an improvement in the calibration model. The obtained preliminary results have shown that the developed LED-based technique can be sufficiently accurate for the analysis of milk fat content. The ways of its further development and improvement have been discussed.

## 1. Introduction

The amount of fat in milk is an important criterion for its quality. Fat is the most valuable component of milk, which greatly affects its price [1]. Therefore, the fat content must be thoroughly controlled in the course of milk collection and processing. Besides, it is a simple marker of the cow’s health. Milk with a fat-to-protein ratio over 1.5 may indicate possible ketosis, i.e., an excess of ketone bodies [2]. Conversely, a low-fat content could be a sign of subacute ruminal acidosis [3], which is a dangerous excess of acids in the animal’s blood.

In accordance with the International Organization for Standardization (ISO), the fat content in milk is determined with chemical methods: the acid method by Gerber [4] and the gravimetric method by Röse-Gottlieb [5]. Chemical methods of analysis have high accuracy, but their implementation is labor- and time-consuming and requires highly qualified personnel. Additionally, dangerous reagents are often used in chemical methods. Spectroscopy in the mid infrared (MIR) region is the most widespread instrumental method of milk analysis. According to ISO 9622:2013 [6], the MIR analyzer in the spectral range of 0.4–11 μm can be used for accurate estimation of mass fractions of fat, proteins, lactose, dry matter, and other parameters of milk quality. The main disadvantage of MIR spectroscopy is the high price of the instrument acquisition and maintenance. As a consequence, such devices are mainly used only in large dairies and research centers.

Optical spectroscopy in the visible and short-wave near infrared (Vis/SW-NIR) region is rarely used in the analysis of dairy products due to the low extinction coefficients of the components in this region (400–1100 nm). However, a number of previous studies have shown that the analysis of milk samples in this region can be fast, non-destructive, sufficiently accurate, and reliable to determine the milk fat and protein content [7,8,9,10,11,12,13]. The analysis in the Vis/SW-NIR region is mainly based on the wavelength-dependent light scattering by colloidal milk particles of different sizes [12,13]. The complex spectral response, in this case, can only be quantified using the methods of multivariate data analysis, also known as chemometrics. While MIR spectroscopy is a well-established laboratory method for routine milk quality control, optical techniques in the Vis/SW-NIR region provide flexible and relatively inexpensive solutions for field and express analysis, for example, in dairy farms.

The scatter-based Vis/SW-NIR spectroscopy suits well for the creation of optical multisensor systems (OMS)—specialized analyzers for various samples and media [14,15,16]. OMS is an analytical device that includes a set of two or more optical sensors (sensor channels) optimized for a particular application [17]. The operation of multisensor devices is based on obtaining and mathematical processing a complex unresolved analytical signal to build a predictive mathematical model, for instance, a calibration model for a mixture component. Among all the platforms proposed to date, optical sensors based on light-emitting diodes (LEDs) [18,19] are the most attractive in terms of creating simple and inexpensive analytical devices.

Nowadays, a lot of research is being carried out to develop compact milk analyzers based on various physical principles [1,20,21,22,23,24,25,26]. Portable analyzers are designed to analyze milk for fat, protein, lactose, and other bulk components. For example, it has been shown that it is possible to measure the fat content of milk in real-time using a W-type fiber optic sensor [20]. There is a study on determining the fat content in milk using a plastic optical fiber sensor [21]. A novel approach using an array of differently sensitive photodetectors realized on a chip [27] has been successfully tested in fat determination in standard samples of raw milk [26]. LEDs in the visible (Vis) spectral range are rarely used in milk analysis. For example, a low-cost portable analyzer for determining the fat content in milk and its dilution with water based on Vis spectroscopy was proposed in [1]. This spectrometer consists of a microcomputer spectral sensor with a single chip that measures the intensity of six LEDs at specific wavelengths. Simplified analyzers based on image analysis [28,29] of diffuse spots of LED-lighted milk have been proposed for fat and protein determination in milk. Despite the growing interest in LED-based optical sensing, in particular in milk analysis, this approach remains an area of active research rather than practical implementation.

The purpose of this work was to develop and test an RGB sensor consisting of three LEDs (red, green, and blue) for the quantitative determination of fat in milk. It was necessary to perform a proof-of-concept of the proposed scatter-based technique. The analysis was deliberately limited to the most practicable spectral region of the visible light (360–780 nm) and implemented in a desktop device using the most common LEDs and a single photometric detector.

## 2. Materials and Methods

### 2.1. LED Sensor Design

The developed analytical device has a modular structure (Figure 1) and works in an automatic mode according to the program built into the Arduino Nano microcontroller based on the ATmega328 chip. The serial port of the microcontroller is connected to the computer with a USB cable for data transfer and for the power supply. The device is powered by a stabilized voltage of 5 V.

Three digital outputs of the controller operate the switching of three LED emitters: red (R), green (G), and blue (B) by means of electronic switches assembled on field-effect transistors, which are part of a powerful RGB LED. The individual LEDs are powered alternately in a pulse mode from a reference current source (100 mA) assembled on the basis of a regulated TL431A stabilizer. The pulse duration is 50 ms.

A photoresistor, powered from the second reference current source (5 μA), serves for the detection of an optical signal. The raw analytical signal is a drop in voltage formed at the photoresistor output. The signal is proportional to the change in resistance caused by the incident light attenuated by interaction with a milk sample. This signal goes to the differential input of an external module of a 16-bit analog-digital converter (ADC). To minimize the power supply noise, the ADC operates from a 4.8 V reference voltage source. Data transfer from the ADC to the controller is carried out via the I2C protocol. The device housing was 3D-printed of non-transparent plastic polylactic acid. It has a slot for placing a cuvette and a cover to protect the sample from ambient light.

### 2.2. Milk Samples

Two sets of milk samples were examined to determine the effect of different fat content on the RGB sensor response.

The first set was prepared by mixing normalized cow’s milk from two packages purchased from a local supermarket with a known fat content of 1.5% and 3.2%. In accordance with the experimental design, the calculated portions of the source milk were mixed with each other to obtain 10 samples with a uniformly distributed fat content: 1.50%, 1.69%, 1.88%, 2.07%, 2.26%, 2.44%, 2.63%, 2.82%, 3.01%, and 3.2%. No additional reference analysis of fat content was performed in this case. It was assumed that the fat content indicated on the milk packaging was accurate enough. Otherwise, some insufficient model bias could be present. But this is not critical for this preliminary study with homogenized milk, the purpose of which is to test individual LED channels and compare sensory and full-spectrum approaches.

The second dataset consisted of 29 raw natural milk samples with varying fat content from 3.10% to 6.01%. These samples were received from the Samara Oblast Agro-Industrial Complex Support Center (Samara, Russia), which monitors animal health and milk quality. The samples were taken from each cow separately during milking. A tablet of Broad Spectrum Microtabs II (Bentley Instruments Inc, Chaska, MN, USA) preservative was added to milk samples to prevent spoilage before they arrived at the laboratory. As a consequence, the samples were pink in color. Each row milk sample was preliminarily analyzed for fat content using a CombiFoss^TM^7 (FOSS Electric, Hilleroed, Denmark) as a reference method. The reference analysis was performed by the Samara Regional Veterinary Laboratory (Samara, Russia).

### 2.3. Spectroscopic and Sensor Measurements

To compare the results obtained on the sensor data with the full spectral method, the milk datasets were additionally measured on a diode-array spectrometer TIDAS E (J&M Analytik AG, Essingen, Germany). This spectrometer was used in our previous research [13,30] to develop a scatter-based method of fat and protein analysis in raw milk. The Vis/SW-NIR spectra in the region 400–1100 nm were acquired in diffuse transmission mode through a 4-mm glass cell (Hellma GmbH, Müllheim, Germany) at room temperature. The integration time was 1 s. The spectra were interpolated to a 1-nm step using a piece-wise linear function. A built-in optical filter that uniformly attenuates the spectrum of incident light by four orders of magnitude was used as a reference. Each milk sample was measured on a spectrometer three times, and the resulting spectra were averaged.

The RGB sensor operates as follows. A milk sample of about 3 mL volume is poured into a transparent cuvette with an optical path length of 1 cm. For the measurement, the cuvette is placed into the slot and covered with a lid from above. The device is connected to a computer using a USB cable, and a serial port monitor is opened on the screen. The data from the photodetector is digitized and filtered by averaging and sent in a tabular form to the port every 100 ms. The output data value is, therefore, proportional to the power of the light flux that is passed through the cuvette with the milk sample. Three detected signals reflect alternating pulses of LEDs irradiation at different wavelengths, as corresponds to the red with an emission maximum of 631 nm and full width at half maximum (FWHM) of 16 nm (i.e., an operating range of 600–654 nm), green with an emission maximum of 531 nm and FWHM of 28 nm (i.e., an operating range of 483–592 nm) and blue with an emission maximum of 466 nm and FWHM of (i.e., an operating range of 425–515 nm) channels (Figure 2) of the developed device.

According to the scheme shown in Figure 3, each sample was measured three times to test the reproducibility of the measurement by taking a new portion of milk. Since the developed analyzer produces the readings continuously, each measurement took about 10 s, and 10 scans were obtained and averaged. These three measurements were then also averaged. Thus, one sample is represented by one measurement in the model.

Signal reproducibility (mean and standard deviation (SD)) for three sensor channels is presented in Table 1. The reprodusibility is demonstrated for two samples of homogenized milk with minimum and maximum fat content (Table 1). The maximum difference between the mean values of the two milk samples was found for the red channel (difference is 854.82) and the minimum for the blue channel (difference is 342.54). The smallest SD is for the red channel.

### 2.4. Data Analysis

The calibration model relating fat content with the optical sensor response for each channel (LEDs) was calculated using a linear regression equation (Equation (1)).
y *=* b_0_ + b_1_x(1)
where y is the sensor response from individual channels, x is the fat content, and b_0_ and b_1_ are bias and slope coefficients, respectively.

To relate the sensor response through two or three channels at once to fat content and to build a predictive model, two chemometric methods, such as projection on latent structures (PLS) [31] and multiple linear regression (MLR) [32], were tested. MLR is used to assess the relationship between two or more independent variables and one dependent variable and works well when there are not many independent variables (fewer than samples), such as in our case, where there are only three variables [33].

The vector of regression coefficients b in MLR is calculated directly from X- and y-variables (Equation (2)):(2)$$b={\left(X^{T}X\right)}^{-1}X^{T}y$$

The PLS algorithm is trained on a dataset of X-variables (spectral intensities or sensor response) and a corresponding y-variable (a priori known fat concentrations), optimizing the regression coefficients in a linear equation, which relates X and y. The optimized coefficients can be used for predicting the value of the y-variable from a previously unseen set of X-variables. Leave-one-out cross-validation (CV) on averaged datasets was applied to estimate the PLS model performance.

The model performance for both regression methods is assessed by two standard metrics—the root-mean-square error (Equation (3)) of calibration (RMSEC) or cross-validation (RMSECV) and the correlation coefficient R^2^ (Equation (4)):(3)$$\mathrm{RMSE}=\sqrt{\frac{\sum_{i=1}^{k}{\left({\hat{y}}_{i}-y_{i}\right)}^{2}}{k}}$$
(4)$$R^{2}=1-\frac{\sum_{i=1}^{k}{\left({\hat{y}}_{i}-y_{i}\right)}^{2}}{\sum_{i=1}^{k}{\left({\hat{y}}_{i}-\overset{–}{y}\right)}^{2}}\;,\;\mathrm{where}\;\overset{–}{y}=\frac{\sum_{i=1}^{k}{\left({\hat{y}}_{i}-y_{i}\right)}^{2}}{k}$$
where y_i_ and ŷ_i_ are known and predicted values, k—is the number of samples in the validation set (for CV, k is the number of samples).

RMSECV shows the error of the model predictions in the range of concentrations being studied. The coefficient of determination R^2^ is a unitless measure of the model quality, which takes values between 0 and 1.

Four evident outliers were excluded from the modeling for a dataset of raw milk samples. The data analysis was carried out in the Unscrambler 9.7 (CAMO, Oslo, Norway) software package.

## 3. Results and Discussion

### 3.1. LED-Sensor Development

The photo of the RGB sensor prototype is shown in Figure 4. The developed analyzer has been assembled from affordable and inexpensive elements. The housing size of 105 × 55 × 38 mm leaves enough place for the electronics and the sample. These dimensions provide stability for the device on the desktop. If necessary, the size of the analyzer can be significantly reduced.

The sample compartment includes two round apertures on opposite sides of the cuvette, allowing the measurement in transmission mode. One aperture (15.5 mm in diameter) focuses the LED light by means of a single secondary lens into the middle of the cuvette volume. The other (5.2 mm in diameter) serves to feed the light after the sample to the detecting photodiode. The distances from the LED source and from the detector to the cuvette were about 0.2 mm. No optics between the cuvette and detector was used in the RGB sensor prototype. Appropriate lenses could be useful to improve the result of analysis, but in this prototype, it was decided to stick to a minimalistic implementation.

The 3 W high-power LED chip lamp with red, green, and blue LEDs was chosen for analysis as the most common LED source in the Vis region. In this case, it is important that the source of light is a point light. This ensures the same conditions for sample analysis at different wavelengths. The pulse duration of 10 ms was chosen to provide a fast averaging measurement without the influence of transient effects.

Powering via a USB cable has several advantages over an external power source. First, it ensures the simplicity and reliability of the design. Secondly, the PC provides a constant stabilized voltage level. In addition, USB provides a sufficiently high level of current (up to 500 mA) to power high-power LEDs.

An optical path of 1 cm was chosen because it is the most common standard in the spectroscopy of the Vis region. It was previously verified that this optical path length is sufficient to obtain an informative signal at the output.

Based on the principle of simplicity chosen for this prototype, a raw detected signal was used in the calibration model building. Alternatively, the sensor response could be expressed in absorption or transmission units, as in spectroscopy. However, there is no ready-to-use standard sample or reference material that mimics the optical properties of milk and can therefore be used for the reference measurement. This problem is typical in the spectroscopic analysis of milk in the Vis and NIR regions. Custom reference materials are usually developed for a particular analytical instrument [11,13]. In the future, a standard sample can be developed for the present RGB sensor, for example, a light-scattering or absorbing filter suitable for the cuvette compartment. Here, it was important to check the possibility of using the raw detected signals of the sensor channels to build calibration models using linear modeling methods.

### 3.2. Building Calibration Models for Homogenized Milk

The averaged sensor responses for ten calibration samples of homogenized milk are shown in Figure 5. The rainbow gradient in the plot (Figure 5) reflects the increasing fat content in milk. The graph shows that the samples lie presumably in the order of increasing fat content for all three channels. The dots are distributed more evenly for the red channel and less evenly for the blue channel.

At the first stage of the analysis, univariate regression models were built to investigate the relationships between the channel responses and the fat content. The dependencies of the channel responses on the fat content in ten homogenized milk samples are shown in Figure 6. The linear regression equations relating the sensor responses to the fat content were calculated using linear least-squares regression (Equation (1)). The model validation statistics are presented in Table 2. The coefficients of determination show that the red channel (R^2^_CV_ = 0.98, Figure 6A) has the highest correlation with the milk fat content. The green channel (R^2^_CV_ = 0.91, Figure 6B) performs slightly worse. Finally, the blue channel (R^2^_CV_ = 0.51, Figure 6C) is the least informative in terms of fat content determination.

The univariate regression model discussed above was useful to give a relative picture of the channel informativeness and, therefore, the importance of the determination of fat content in milk. In the next step of research, multivariate regression models were built. Two algorithms of chemometrics were used and compared: PLS and MLR. PLS was chosen as currently the most widespread regression algorithm for the analysis of spectral and similar data [31]. On the contrary, MLR is rarely used in spectroscopy because it requires the number of samples to be greater than the number of responses, which is a serious limitation for spectral data analysis. Having only three responses, i.e., sensing channels in our case, releases this limitation and makes the simple and straightforward MLR algorithm fully competitive. The algorithm comparison statistics are presented in Table 2. This comparison was made for different channel combinations, including single channels, the most promising pairwise combination thereof (as follows from the above univariate models), and the full data of three channels together.

PLS and MLR models were also built for individual sensor channels to compare the results of univariate regression analysis with multivariate regression. The combination of the three channels gives significantly better validation statistics of the PLS model (RMSECV = 0.08 and R^2^ = 0.98) as corresponds to the predicted vs. reference plot in Figure 7A compared to both the univariate models and the red and green channel combination. The scores and loadings plots of the PLS model are shown in Figure 7B,C, respectively. The PLS scores (Figure 7B) show that the sample order along the first latent variable (LV1) generally follows the increasing fat content. The loadings in Figure 7C demonstrate the relative contribution of each sensor channel to the model. Thus, the red channel was found to be the most informative, and the blue channel was the least informative, which is in agreement with the univariate analysis results.

The validation statistics of MLR and PLS regression are very similar (Table 2). This similarity of results for the two algorithms confirms our previous statements [34] that MLR can be a preferred regression method for building calibration models on multisensory data, while PLS suits perfectly for the conventional spectra.

Table 2 also presents the data from full-spectrum analysis performed on the TIDAS E spectrometer (Section 2.3). As expected, the spectroscopic technique shows a noticeably better model performance for the same set of calibration samples. However, the accuracy of determination of fat content with the developed prototype is still acceptable in many practical applications in dairy.

### 3.3. Building Calibration Models for Raw Milk

Twenty-nine raw milk samples were analyzed using the developed RGB sensor, and a PLS model was built (Figure 8). As in the case of homogenized milk (Section 3.2), the PLS regression statistics for the full RGB-sensor data model (R^2^_CV_ = 0.72, RMSECV = 0.37) is significantly better than for individual sensor channels or combinations thereof. It is slightly worse than the model built on the full-spectrum data (Figure 9B) of the same samples (R^2^_CV_ = 0.76, RMSECV = 0.35). In general, the model performance in the case of raw milk data is moderate and can only be used in less critical practical applications. This result has a straightforward explanation. First of all, it is related to a wide variability of fat globule sizes in natural raw milk, which add the light scattering dependence on the wavelength [13].

An additional complication was due to the presence of a preservative added to each sample (hence the pink color of the milk). Even though the quantity of added preservatives was the same, due to the difference in sample volume, its actual concentration was variable, which could be seen from the spectra (Figure 9B) and from the sample color itself. In a multivariate sense, the addition of a preservative to the samples adds one more source of variability to the spectral and sensory data. Although its concentration is unknown, its effect can be correctly accounted for by adding an additional factor (i.e., a latent variable) to a multivariate regression model, such as PLS. The pink color of the samples indicates the main absorption in the short-wave part of the spectra. Indeed, one can see this absorption in full-spectrum data obtained using the TIDAS E instrument (compare Figure 9A,B), and the corresponding spectral variance is moderate compared to the variance due to the milk composition difference. It can also be seen that the preservative has the most affected on the blue LED channel (compare Figure 2 and Figure 9). However, this channel contributes less to the model than the others, as shown in the homogenized milk study (Table 2). Perhaps, the presence of an additional variability factor in the already complex spectral response of milk is one of the reasons for the insufficient performance of the PLS model in the raw milk case, even for the full-spectrum data. In our previous study [13], five factors (LVs) were required to build a well-performing PLS model since other components of milk, such as protein and lactose, had a variable content and light-scattering properties. In this case, one can expect a similar or even higher complexity of the system.

In the sensory approach, the number of LEDs forming the sensor variables (channels) should be at least as large as the number of LVs, even for their optimal configuration [35]. Also, the dataset of 29 historical samples (no experimental design was used) is hardly sufficient to train the calibration model appropriately. At such a high level of complexity, many more samples may be required to adequately train the calibration model [30]. However, it was not the aim of the present study with the sensor prototype.

Therefore, we came to the conclusion that the present version of the LED-based sensor does not suit well for the determination of fat in raw milk samples. In order to fit it to the real system complexity, a few LEDs in the Vis and NIR range should be added in an optimal way. With this extension, the original concept of the RGB sensor as a simple solution for fat content determination should be revised. Nevertheless, this approach may work well for homogenized milk, the multivariate complexity of which is significantly lower [36].

Overall, the reported results show that the proposed sensory approach, based on a very common combination of LEDs in the visible light region, is suitable for the analysis of fat content in homogenized milk, which is a staple of the dairy industry. The method using the powerful RGB LED as a working element can be further improved. The fat content prediction accuracy can be further enhanced by technical optimization, i.e., the inclusion of focusing and collimating optics, optical path, and LED intensity adjustment, etc. In the case of raw milk, however, the spectral range of analysis should probably be extended towards the NIR range, and the number of LEDs should be increased in accordance with the real complexity of samples. This should be the main effort of further research.

## 4. Conclusions

The obtained results are a step forward in the development of LED-based sensors for the quantitative analysis of milk composition. The possibility of creating such a system for determining milk fat, theoretically shown earlier [35], was experimentally proven using the developed desktop prototype presented here. It has been shown that responses of individual LEDs in the Vis light region have very different correlation strengths with the fat content and are generally not suitable for its accurate determination. Combining multiple optical sensors at different wavelengths in the same measurement improves prediction accuracy. Therefore, multisensory measurement is necessary to take into account various factors that affect the result of light transmission through a milk sample, in particular, differences in the number and size of colloidal milk particles.

Deliberate simplicity was the main constructional principle put into the basis of the developed RGB sensor, which uses only widely available low-cost components, 3D-printed parts and avoids any precision optics. Nevertheless, this proof-of-concept study has shown that the proposed design of the OMS is suitable for determining fat content in homogenized milk, which is the most produced and consumed dairy product. The accuracy of prediction is comparable to the well-established spectral approach. Measurements on raw milk samples gave a less accurate model, which is expected given the greater complexity of the system. This experiment has shown that three LEDs are not enough in this case, and for a successful analysis of natural non-homogenized milk, the system must be optimized in terms of the number of LEDs and their operating wavelengths.

Simultaneous determination of proteins and possibly other components having scattering and absorption in the selected region could be a very important extension of the future optimized technique. If necessary, the spectral region can be extended towards the NIR and even the MIR range.

## Figures and Tables

**Figure 1 sensors-23-06861-f001:**
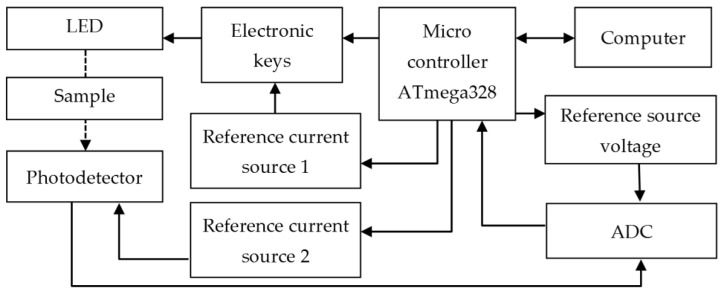
Block diagram of RGB sensor architecture.

**Figure 2 sensors-23-06861-f002:**
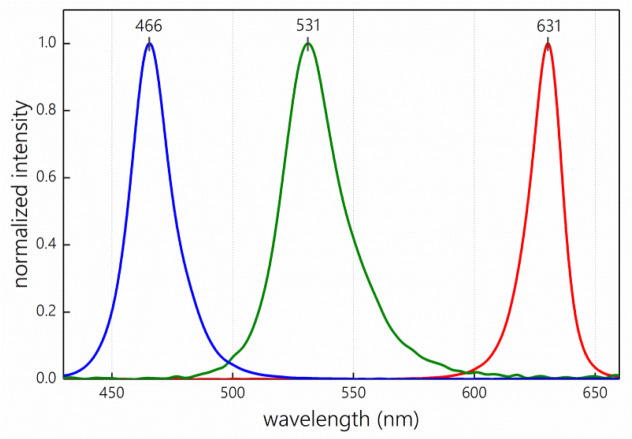
Emission spectra of RGB sensor. LEDs normalized to unit intensity.

**Figure 3 sensors-23-06861-f003:**
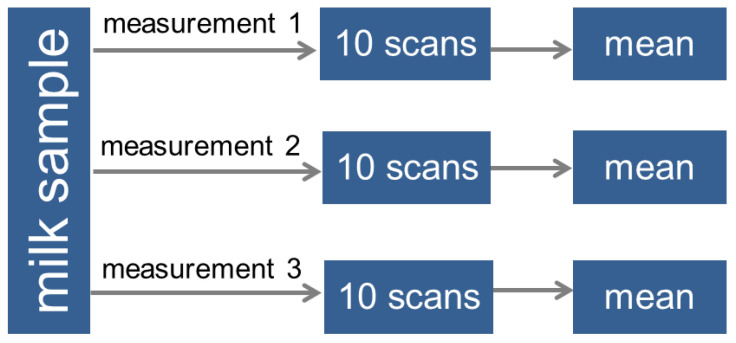
Scheme of primary data processing in the analysis of a milk sample.

**Figure 4 sensors-23-06861-f004:**
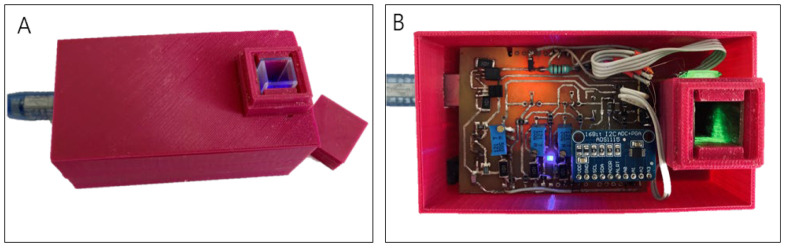
Photo of the sensor prototype: (**A**) sensor with cuvette and measurement compartment lid; and (**B**) internal view of sensor without top cover.

**Figure 5 sensors-23-06861-f005:**
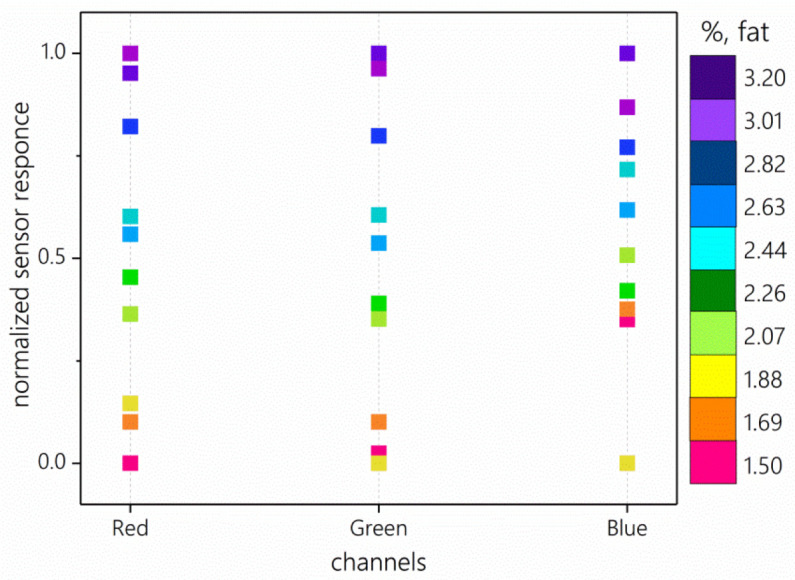
Normalized sensor responses for ten samples of homogenized milk. The fat content in the samples is indicated using the color scale on the right side of the figure.

**Figure 6 sensors-23-06861-f006:**
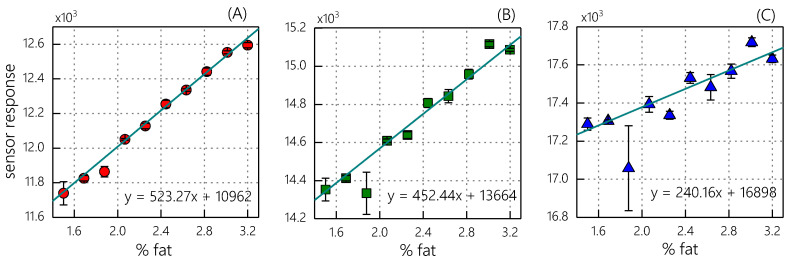
Observed responses of individual sensor channels versus fat content in homogenized milk samples: (**A**) red; (**B**) blue; and (**C**) green LED channels. The regression lines and the corresponding linear regression equations are presented on the respective graphs. The error bars represent the standard deviation of three replicated measurements.

**Figure 7 sensors-23-06861-f007:**
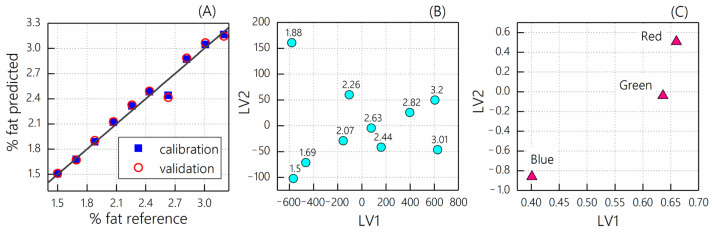
PLS regression results for homogenized milk samples: (**A**) predicted versus reference plot for fat content determination; (**B**) scores plot; (**C**) loadings plot.

**Figure 8 sensors-23-06861-f008:**
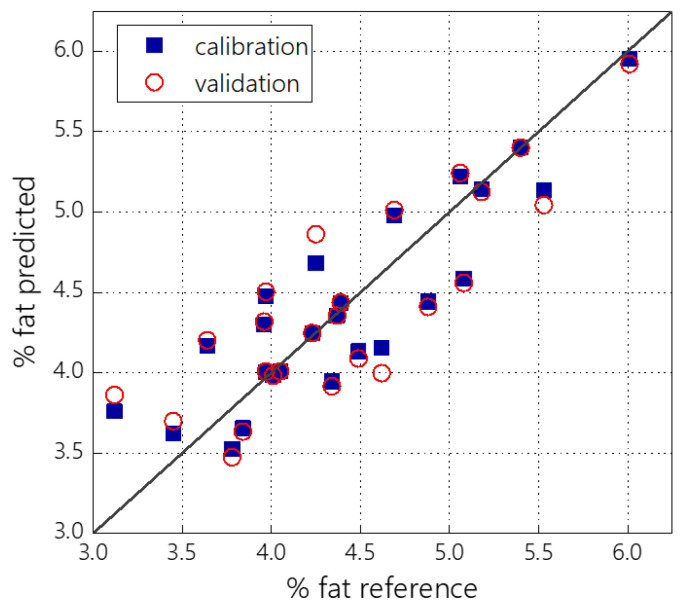
PLS-predicted versus reference plot for fat content in raw milk samples.

**Figure 9 sensors-23-06861-f009:**
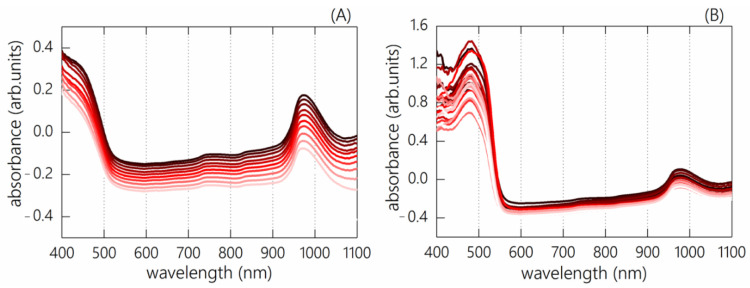
Spectra of homogenized (**A**) and raw (**B**) milk samples obtained using TIDAS E spectrometer. Fat content is coded by a red color gradient: a darker color corresponds to a higher fat concentration.

**Table 1 sensors-23-06861-t001:** Signal reproducibility of each sensor channel for homogenized milk with the minimum and maximum fat content.

Fat, %	Channel ^1^	Mean ^2^	SD Range for Scans
1.50	R	11,739.36	7.22–12.99
	G	14,352.55	7.81–17.27
	B	17,289.61	15.21–22.40
3.20	R	12,594.18	5.11–17.78
	G	15,086.58	13.02–20.35
	B	17,632.15	10.91–13.66

^1^ R—red, G—green, B—blue; ^2^ mean for three measurements.

**Table 2 sensors-23-06861-t002:** Calibration and validation statistics for PLS and MLR regression models for fat content prediction in homogenized milk.

Dataset ^1^	LV	Calibration		CV	
		RMSE ^3^, %	R^2^	RMSE, %	R^2^
PLS					
R	1	0.08	0.98	0.10	0.98
G	1	0.14	0.94	0.18	0.91
B	1	0.29	0.71	0.42	0.51
RG	1	0.10	0.96	0.13	0.95
RGB	2	0.07	0.98	0.08	0.98
Spectrometer ^2^	2	0.02	1.00	0.03	0.99
MLR					
R	-	0.08	0.98	0.10	0.98
G	-	0.14	0.94	0.18	0.89
B	-	0.29	0.71	0.42	0.51
RG	-	0.07	0.98	0.08	0.98
RGB	-	0.07	0.99	0.10	0.97

^1^ Calibration datasets used for the model building were composed of individual channel responses and their combinations: R—red, G—green, B—blue, RG—red and green, RGB—red, green, and blue; ^2^ calibration dataset composed of full spectra of milk samples in the wavelength range 400–1100 nm obtained using TIDAS E spectrometer; ^3^ concentration range 1.5–3.2% of fat.

## Data Availability

Not applicable.

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
