# Peer review of "LED-Based Desktop Analyzer for Fat Content Determination in Milk"

_sensors, 2023, doi:10.3390/s23156861_

Round 1

Reviewer 1 Report

In this manuscript (sensors-2479978), The authors developed and tested an RGB sensor consisting of three LEDs for the quantitative determination of fat in milk. The performance of this method is interesting and satisfying, and it has great novelty. However, there are still some areas of the article that need to be optimized. Thus, I would like to recommend this manuscript to be published in sensors after major revisions.

Here are some reasons why we give this opinion:

1.     For the accuracy of the data, the milk samples purchased from the local supermarket in 2.2 need to be further tested for fat content.

2.     It is recommended that the results of the experiment be presented more in the form of data rather than in words. For example, in Fig. 3, it is recommended that the reproducibility of the method be demonstrated in the form of data.

3.     Data obtained from multiple measurements of the same sample and from measurements of different samples which contain the same fat content should be presented, together with the RSD values of the method for these repeated measurements.

In this manuscript (sensors-2479978), The authors developed and tested an RGB sensor consisting of three LEDs for the quantitative determination of fat in milk. The performance of this method is interesting and satisfying, and it has great novelty. However, there are still some areas of the article that need to be optimized. Thus, I would like to recommend this manuscript to be published in sensors after major revisions.

Here are some reasons why we give this opinion:

1.     For the accuracy of the data, the milk samples purchased from the local supermarket in 2.2 need to be further tested for fat content.

2.     It is recommended that the results of the experiment be presented more in the form of data rather than in words. For example, in Fig. 3, it is recommended that the reproducibility of the method be demonstrated in the form of data.

3.     Data obtained from multiple measurements of the same sample and from measurements of different samples which contain the same fat content should be presented, together with the RSD values of the method for these repeated measurements.

Author Response

  1. For the accuracy of the data, the milk samples purchased from the local supermarket in 2.2 need to be further tested for fat content.

Authors: The reviewer is conceptually right. The calibration samples used to build and validate calibration models should be analyzed for fat content using any accurate reference method, such as Röse-Gottlib or one of the IR-spectroscopy-based FOSS instruments. However, in this particular case of our proof-of-concept study this requirement can be relaxed, and the target fat percentage indicated at the milk package can be taken as a reference for modeling. We assume that milk producers have strong interest in determining this value as exact as possible using appropriate laboratory methods, since the food quality it is strictly controlled by the state. But if it is still inaccurate, the error can only worsen the model statistics, and therefore, represents an additional challenge for the method test. Some insufficient bias because of the dilution method used, is also not critical for the conclusion on how good the linearity of response and therefore the accuracy of the sensor can be. Therefore, we use this simplified approach in our preliminary and feasibility studies. It was successfully tested before using a much larger set of market milk samples [36]. This approach and motivation have been better described in section 2.2, in particular: ”It was assumed that fat content indicated on the milk packaging was accurate enough. Otherwise, some insufficient model bias could be present. But this is not critical for this preliminary study with homogenized milk, the purpose of which is to test individual LED channels and compare sensory and full-spectrum approaches.”

[36]      Surkova, A.; Belikova, V.; Kirsanov, D.; Legin, A.; Bogomolov, A. Towards an optical multisensor system for dairy: Global calibration for fat analysis in homogenized milk. Microchem. J. 2019, 149, 104012. https://doi.org/10.1016/j.microc.2019.104012

  1. It is recommended that the results of the experiment be presented more in the form of data rather than in words. For example, in Fig. 3, it is recommended that the reproducibility of the method be demonstrated in the form of data.

Authors: We have added Table 1 with mean signal values as well as SD for each sensor channel.  Additionally, we have added the error bar representing the standard deviation of three replicated measurements on Figure 6.

  1. Data obtained from multiple measurements of the same sample and from measurements of different samples which contain the same fat content should be presented, together with the RSD values of the method for these repeated measurements.

Authors: We have replied this question in the previous answer.

Reviewer 2 Report

The manuscript (sensors-2479978) demonstrates that a portable desktop analyser, using light scattering from three LEDs, can accurately determine fat content in cow's milk, with combined LED signals improving calibration. This cost-effective, high-throughput method offers potential for further development and enhancement.

However, the manuscript is not adequate or publication. Basic elements are missing in all sections, from the introduction to the conclusions

The captions for the figures require better descriptions of all the figures and tables present. In addition, there is no mention of a discussion section. The results are written in a simple manner.

The manuscript is not adequate. Basic elements are missing from the captions that do not reflect what they mean. The discussions also do not appropriately reflect the characteristics and the new discoveries. For example, what are the actual samples analysed? Why were no comparative analyses conducted with other spectrometers?

Keywords in alphabetic order;

L52. “Vis/SW-NIR” to “Vis/NIR-SWIR”; Check all manuscript;

The manuscript also does not follow the standard reported in the Instructions for Authors. Additionally, some references are outdated and need to be updated.

Figure 6 not adequate what is A and B?

Check old references.

English need improve.

Author Response

The captions for the figures require better descriptions of all the figures and tables present. In addition, there is no mention of a discussion section. The results are written in a simple manner.

Authors: All figure and table captions, descriptions and notes have been improved. As well, the text in the Results and discussion section (formerly Results) has been extensively revised. In particular:

The following explanation was added in subsection 3.1:

“Based on the principle of simplicity chosen for this prototype, a raw detected signal was used in the calibration model building. Alternatively, the sensor response could be expressed in absorption or transmission units, as in spectroscopy. However, there is no ready-to-use standard sample or reference material that mimics the optical properties of milk and can therefore be used for the reference measurement. This problem is typical in the spectroscopic analysis of milk in the visible and near infrared region. Custom reference materials are usually developed for a particular analytical instrument [11,13]. In the future, a standard sample can be developed for the present RGB-sensor, for example, a light-scattering or absorbing filter suitable for the cuvette compartment. Here, it was important to check the possibility of using the raw detected signals of the sensor channels to build calibration models using linear modelling methods.”

[11]      Kalinin, A.V.; Krasheninnikov, V.N.; Krivtsun, V.M. Short-wave near infrared spectrometry of back scattering and transmission of light by milk for multi-component analysis. J. Near Infrared Spectrosc. 2013, 21, 35–41. https://doi.org/10.1255/jnirs.1034 

[13] Bogomolov, A.; Melenteva, A. Scatter-based quantitative spectroscopic analysis of milk fat and total protein in the region 400-1100 nm in the presence of fat globule size variability. Chemom. Intell. Lab. Syst. 2013, 126, 129–139. https://doi.org/10.1016/j.chemolab.2013.02.00

The manuscript is not adequate. Basic elements are missing from the captions that do not reflect what they mean. The discussions also do not appropriately reflect the characteristics and the new discoveries. For example, what are the actual samples analysed? Why were no comparative analyses conducted with other spectrometers?

Authors: In addition to figure and table captions, some section titles were improved. The entire text of section 3 (Results and discussion) and 4 (Conclusions) has been significantly improved to better outline the novelty of the results and to make the descriptions more complete.

Keywords in alphabetic order;

We checked the journal requirements and there is no requirement for alphabetical order. The requirements are just that:

“Keywords: Three to ten pertinent keywords need to be added after the abstract. We recommend that the keywords are specific to the article, yet reasonably common within the subject discipline.”

We follow the logic that keywords are arranged in order from more general terms to specific terms.

L52. “Vis/SW-NIR” to “Vis/NIR-SWIR”; Check all manuscript;

There is no standard abbreviation for the short-wave near-infrared region (780-1100 nm) in the literature. We stick to the abbreviation SW-NIR, as in our previous papers [13,30,36], to distinguish region of 780-1100 nm from the full NIR region.

[13] Bogomolov, A.; Melenteva, A. Scatter-based quantitative spectroscopic analysis of milk fat and total protein in the region 400-1100 nm in the presence of fat globule size variability. Chemom. Intell. Lab. Syst. 2013, 126, 129–139. https://doi.org/10.1016/j.chemolab.2013.02.00

[30] Melenteva, A.; Galyanin, V.; Savenkova, E.; Bogomolov, A. Building global models for fat and total protein content in raw milk based on historical spectroscopic data in the visible and short-wave near infrared range. Food Chem. 2016, 203, 190–198. https://doi.org/10.1016/j.foodchem.2016.01.127

The manuscript also does not follow the standard reported in the Instructions for Authors. Additionally, some references are outdated and need to be updated.

Authors: We have checked the Instruction for Authors and introduced the necessary improvements in the manuscript.

Figure 6 not adequate what is A and B?

Authors: The missing captions for Figure 6B and 6C were added.

Check old references.

Authors: The old references were updated.

Reviewer 3 Report

The manuscript reports an innovative LED-based analyzer for fat content determination in milk. The application is interesting and could be a new tool in the field.

There are however two main critical issues:

- authors do not report if the fat % of the  two purchased packages of normalized cow’s milk has been determined. This is fundamental since the milk of the two packages, mixed in different ratios, is shown as the reference;

- as described by the authors, the real milk (received from the Samara Oblast Agro Industrial Complex Support Center) shows a light pink color due to the addition of a tablet preservative to prevent the milk from spoiling before the delivery at the lab.  This, in my opinion, does not allow to understand if the method can be sensible,  robust and reproducible or if the different color due to the addition is too much interfering on the results.

I suggest a major revision, mainly on this last critical issue.

Moderate editing of English language is required along the text

Author Response

There are however two main critical issues:

- authors do not report if the fat % of the two purchased packages of normalized cow’s milk has been determined. This is fundamental since the milk of the two packages, mixed in different ratios, is shown as the reference;

Authors: The reviewer is conceptually right. The calibration samples used to build and validate calibration models should be analyzed for fat content using an accurate reference method. However, in this particular case of our proof-of-concept study this requirement can be relaxed, and the target fat percentage indicated at the milk package can be taken as a reference for modeling. We assume that milk producers have strong interest in determining this value as exact as possible using appropriate laboratory methods, since the food quality it is strictly controlled by the state. But if it is still inaccurate, the error can only worsen the model statistics, and therefore, represents an additional challenge for the method test. Some insufficient bias because of the dilution method used, is also not critical for the conclusion on how good the linearity of response and therefore the accuracy of the sensor can be. Therefore, we use this simplified approach in our preliminary and feasibility studies. It was successfully tested before using a much larger set of market milk samples [36]. This approach and motivation have been better described in section 2.2, in particular: “It was assumed that fat content indicated on the milk packaging was accurate enough. Otherwise, some insufficient model bias could be present. But this is not critical for this preliminary study with homogenized milk, the purpose of which is to test individual LED channels and compare sensory and full-spectrum approaches.”

[36]      Surkova, A.; Belikova, V.; Kirsanov, D.; Legin, A.; Bogomolov, A. Towards an optical multisensor system for dairy: Global calibration for fat analysis in homogenized milk. Microchem. J. 2019, 149, 104012. https://doi.org/10.1016/j.microc.2019.104012

- as described by the authors, the real milk (received from the Samara Oblast Agro Industrial Complex Support Center) shows a light pink color due to the addition of a tablet preservative to prevent the milk from spoiling before the delivery at the lab.  This, in my opinion, does not allow to understand if the method can be sensible, robust and reproducible or if the different color due to the addition is too much interfering on the results.

Authors: In a multivariate sense the addition of a preservative to the samples adds one more source of variability to the spectral and sensory data. Although its concentration is unknown, its effect can be correctly taken into account by adding an additional factor (i.e. latent variable) to the multivariate regression model, such as PLS. Pink color of samples points at the main absorbance in the short-wave part of the spectra. Indeed, one can see this absorbance in full-spectrum data obtained using TIDAS E instrument, and the corresponding spectral variance is moderate compared to the variance due to the milk difference. It can be seen that blue LED channel is the most affected by the preservative. However, this channel contributes less than the others into the model, as it was shown in the study with homogenized milk. Perhaps, the presence of an additional variability factor in an already complex spectral response of milk is one of the reasons for an insufficient performance of the PLS model in the raw milk case, even with the full-spectrum data. In our previous study [13,30], five factors (LVs) were necessary to build a well-performing PLS model. One can expect similar or even higher system complexity in this study. In a sensory approach the number of LEDs forming the sensor variables (channels) should be at least as large as the number of LVs, even for their optimal configuration [35]. Also, the dataset of 29 historical samples (no experimental design was used) is hardly sufficient to train the calibration model appropriately.

Therefore, we came to the conclusion that the present version of the LED-based sensor does not suit for the determination of fat in raw milk samples. To adjust it to the real system complexity a few LEDs in the visible and NIR-range should be added in an optimal way. With this extension, the initial concept of RGB-sensor as a simple solution for fat content determination should be revised. Nevertheless, this approach can be well suited for the homogenized milk, multivariate complexity of which is significantly lower [36].

Similar explanation has been integrated into section 3 of the manuscript.

[13]      Bogomolov, A.; Melenteva, A. Scatter-based quantitative spectroscopic analysis of milk fat and total protein in the region 400-1100 nm in the presence of fat globule size variability. Chemom. Intell. Lab. Syst. 2013, 126, 129–139. https://doi.org/10.1016/j.chemolab.2013.02.00

[30]      Melenteva, A.; Galyanin, V.; Savenkova, E.; Bogomolov, A. Building global models for fat and total protein content in raw milk based on historical spectroscopic data in the visible and short-wave near infrared range. Food Chem. 2016, 203, 190–198. https://doi.org/10.1016/j.foodchem.2016.01.127

[35]      Galyanin, V.; Melenteva, A.; Bogomolov, A.  Selecting optimal wavelength intervals for an optical sensor: A case study of milk fat and total protein analysis in the region 400–1100 nm. Sens. Actuators B Chem. 2016, 218, 97–104. https://doi.org/10.1016/j.snb.2015.03.101

I suggest a major revision, mainly on this last critical issue.

Moderate editing of English language is required along the text

Authors: The manuscript language has been improved for gramma and style.

Round 2

Reviewer 1 Report

The revised manuscript (sensors-2479978) has been improved. For the questions about this work, the authors have made careful revisions and supplements, and the responses are reasonable. Thus, I would like to recommend this paper to be published in sensors.

Reviewer 2 Report

I believe the authors made significant changes and addressed points that were not clear in the manuscript. Now, I think it would be acceptable for publication.

Minor changes.

Reviewer 3 Report

After reading the considerations of the authors in response to the comments, given the changes made, the submission can be accepted in my opinion